# Monitoring climate change impacts, Indigenous livelihoods and adaptation: Perspectives from Inuit community of Hopedale, Nunatsiavut, Canada

## Research Article

climate change; climate adaptation; human impact; climate resilience; coastal adaptation

**Corresponding author:**
Ishfaq Hussain Malik;
Email: i.h.malik@leeds.ac.uk

Ishfaq Hussain Malik[1] [iD], James D. Ford[1,2], Ian Winters[3,4], Beverly Hunter[3,4], Nicholas Flowers[3,5], Duncan Quincey[1], Kevin Flowers[3], Marjorie Flowers[3,6], Dean Coombs[3], Christine Foltz-Vincent[3], Nicholas E. Barrand[7] and Robert G. Way[8]

[1]School of Geography, University of Leeds, Leeds, UK; [2]Priestley Centre for Climate Futures, University of Leeds, Leeds, UK; [3]Hopedale Community Member, Nunatsiavut, Canada; [4]Nunatsiavut Government, Hopedale, Canada; [5]Inotsiavik Language and Culture Inc. Hopedale, Nunatsiavut, Canada; [6]Hopedale Inuit Community Government, Nunatsiavut, Canada; [7]School of Geography, Earth and Environmental Sciences, University of Birmingham, Edgbaston, Birmingham, UK and [8]Northern Environmental Geoscience Laboratory, Department of Geography and Planning, Queen's University, Ontario, Canada

## Abstract

The Arctic is at the forefront of climate change, undergoing some of the most rapid environmental transformations globally. Here, we examine the impacts of climate change on the livelihoods in the coastal Inuit community of Hopedale, Nunatsiavut, Canada. The study examines recently evolved adaptation strategies employed by Inuit and the challenges to these adaptations. We document changing sea ice patterns, changing weather patterns and the impact of invasive species on food resources and the environment. Utilising knowledge co-production and drawing upon Indigenous knowledge, we monitor the changes and multiple stresses through direct observations, engagement with rights holders and community experiences to characterise climate risks and associated changes affecting livelihoods. We use both decolonising research and participatory methodologies to develop collaboration and partnership, ensuring that monitoring reflects local priorities and realities while also fostering trust and collaboration. We showcase that monitoring environmental trends involves more than data collection; it includes observing and analysing how environmental changes affect community well-being, particularly in terms of food security, cultural practices, economic activities, mental health, sea ice changes and weather patterns. The paper contributes to a nuanced understanding of Inuit resilience and experiences in confronting climate risks and the broader implications for Indigenous communities confronting climate challenges.

## Impact statement

The Arctic is experiencing wide-ranging impacts of climate change and is warming nearly four times faster than the global average. These changes are causing disproportionate impacts on Indigenous Peoples' livelihoods, particularly affecting Inuit communities depending on traditional activities such as hunting, fishing and gathering. Inuit in the Arctic have applied different adaptation strategies to cope with these environmental transformations. Monitoring these changes and responses is essential for understanding how climate risks are shaping the lives of Inuit and how adaptation processes evolve over time. This is one of the first studies that monitor the challenges posed by climate change on sea ice conditions, seasonal shifts, food security and livelihoods of Inuit in Hopedale, Nunatsiavut in the Canadian Arctic. It documents the adaptation strategies employed and the challenges facing these adaptations. This study uses decolonising research methods and partners with Inuit communities to co-produce knowledge and utilise traditional ecological knowledge (TEK) and participatory monitoring to monitor, characterise and understand environmental transformations. This study provides a detailed analysis of how Inuit in Hopedale are experiencing different impacts of climate change, the impact of invasive species on food sources, challenges in wood gathering, economic implications, inequity and mental well-being, and the role of sharing networks. It highlights the role of community-led monitoring and local government initiatives in adaptation. It examines the critical role of TEK and the resilience of Inuit communities in adapting to climate change. This study contributes to a deeper understanding of how Indigenous knowledge informs and strengthens monitoring and climate adaptation policies, fostering long-term resilience in the face of global environmental change.



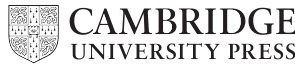

## Introduction

*"I measured sea ice thickness two years ago on 2nd May at Hopedale, and it was 2.5 m thick. Last year, I did it on 5th May, and it was 0.76 m thick. So, it is unpredictable each year. In the last 5 years, super dramatic, pretty big changes have happened. It is perhaps going to be quicker this year. The top crust is melting before cracks in the ice open up so that the water could drain off. As a result, the water (and sun reflecting on the water) could warm up and eat away at the ice below, much faster. No snow to pack down means less insulation for the ice and creates a quicker ice melt." ~ Inuit community member from Hopedale, 2024.*

The Arctic is warming nearly four times faster than the global average, affecting Indigenous Peoples' livelihoods, and their social and cultural activities (Rantanen et al., 2022; Ford et al., 2021), with the potential to extend impacts beyond the Arctic (Mosoni et al., 2024). Indigenous Peoples in Canada (Inuit) living in Inuit Nunangat (the Inuit homeland) are affected by climatic change resulting from rising temperatures, thawing permafrost, reduced sea ice, sea level rise, coastal erosion and storms, causing temporary immobility, relocation and unsafe travel on sea ice (Ayeb-Karlsson et al., 2024; IPCC, 2023). Sea, lake, and river ice are crucial for transportation in Arctic Canada, with 8,000 km of winter ice roads connecting Indigenous communities (Barrette et al., 2022). Climate change affects these routes, resulting in fewer operational travel days, negatively impacting livelihoods and causing adaptation challenges (Culpepper et al., 2024; Solovyeva, 2024; Ford et al., 2023).

The Arctic is experiencing transformative environmental changes with implications for ecosystems and the Indigenous communities reliant on them (Pearson et al., 2023). Monitoring is crucial for tracking these environmental changes and advancing understanding of implications for livelihoods and adaptation, providing a strong base for knowledge co-production, community-based participatory research and meaningful partnerships with communities (Ford and Pearce, 2012; Reiersen et al., 2024; Bishop et al., 2022). More generally, monitoring is essential globally for understanding the complexities and socio-economic impacts of climate change, how climate risks evolve over time and the dynamics of how people experience and respond to climate change (Malik and Ford, 2024a). It is important for understanding the dynamic and multiscale nature of climate change, characterising how climate interacts with community livelihoods including hunting and travelling and associated societal implications (Ford et al., 2013). Beyond the biophysical effects of climate change, monitoring can facilitate the examination of the challenges Inuit face when engaging in subsistence activities and how changing ice, land and ocean habitats intersect with multiple stressors to affect food security and livelihoods (Naylor et al., 2021).

Ecological monitoring is also essential in the Arctic. Long-term projects like the Arctic Biodiversity Assessment (ABA), Circumpolar Biodiversity Monitoring Program (CBMP) and the Arctic Monitoring and Assessment Program (AMAP) provide baseline data for understanding changes in ice cover, seasonal cycles and species distributions. These projects enable rapid detection, prediction, understanding and response to ecological changes and monitor coastal, marine, freshwater and terrestrial ecosystems, extreme events and thresholds, microplastics, pollution, air quality and climate change impacts (CAFF, 2017, 2021; AMAP, 2019, 2021). Many recent studies underscore the value of Indigenous knowledge for real-time ecological insights (Houde et al., 2022; Kaiser et al., 2019; Johnson et al., 2016; Malik, 2024; Manrique et al., 2018; Little

et al., 2023; Hauser et al. 2023; Malik and Ford, 2024b; Dubos et al., 2023; Turner et al., 2022).

Inuit livelihoods in Nunatsiavut, a self-governing territory in northern Labrador with five Inuit communities, have long been connected to the ocean, sea ice, land and subsistence hunting (Brice-Bennett et al., 1977; 2023). Nunatsiavut is the first Inuit region in Canada to achieve formal self-governance in 2005, with the signing of the Labrador Inuit Land Claims Agreement (Labrador Inuit Association, 2005). Inuit have historically relied on a subsistence-based lifestyle, including hunting, fishing and gathering, which are deeply embedded in their social and cultural identity and well-being (ITK, 2021). This connection remains vital today in the face of climate change (Laver et al., 2024; Hancock et al., 2022). Nunatsiavut is experiencing substantial climate change impacts as evidenced by the findings of a reduction in snow and ice cover (Brown et al., 2012; Barrand et al., 2017), permafrost thaw and landscape change and ecological and ethnobotany studies, including Inuit knowledge studies by Nunatsiavut Government (2024), Rapinski et al. (2018), Norton et al. (2021), Barrette et al. (2020), Davis et al. (2021), Fleming et al. (2012), Fleming (2009), and Wang et al. (2024).

While numerous studies have explored the ways of knowing and learning about the land, recognition and naming of landscape features and habitats, wildlife management and impacts of climate change on livelihoods, travel, hunting and mental health in five Inuit communities of Nunatsiavut (Sawatzky et al., 2021; Zurba et al., 2022; Snook et al., 2020; Cunsolo Willox et al., 2013; Procter and Natcher, 2012; Cuerrier et al., 2022), there is limited work focussing specifically on climate change impacts and adaptation in Hopedale (Fleming et al., 2012; Fleming, 2009).

This study aims to fill this research gap by monitoring the impacts of climate change on the livelihoods of Inuit community members in Hopedale, Nunatsiavut. The three objectives of this study are: (1) to assess the key livelihood components affected by climate change in Hopedale, identifying the most susceptible sectors and practices, (2) to analyse the nature and extent of changes experienced by community members due to shifting climatic conditions through real-time human–environment interactions and community-based monitoring and (3) to document and examine adaptation strategies employed by Inuit community members in response to climate-induced changes, including emerging strategies shaped by socio-economic and environmental factors, through collaborative research and knowledge co-production.

## Methodology

### Study area

Hopedale is a coastal community in the self-governing Inuit region of Nunatsiavut (meaning "our beautiful land" in Inuttitut), which is one of the four regions of Inuit Nunangat. It is located (55° 27′ N, 60° 13′ W) on the eastern coast of Northern Labrador in the Atlantic Sea in the province of Newfoundland and Labrador (Figure 1). Inuit in Hopedale speak English and Inuktitut/Inuttitut (Labrador Inuttitut dialect of Inuktitut), and the Inuktitut name of Hopedale is Agvitok, meaning "place of bowhead whales." It is the second largest and second-northernmost Inuit community (from north to south – Nain, Hopedale, Makkovik, Postville and Rigolet) in Nunatsiavut with about 600 residents and is the legislative capital of

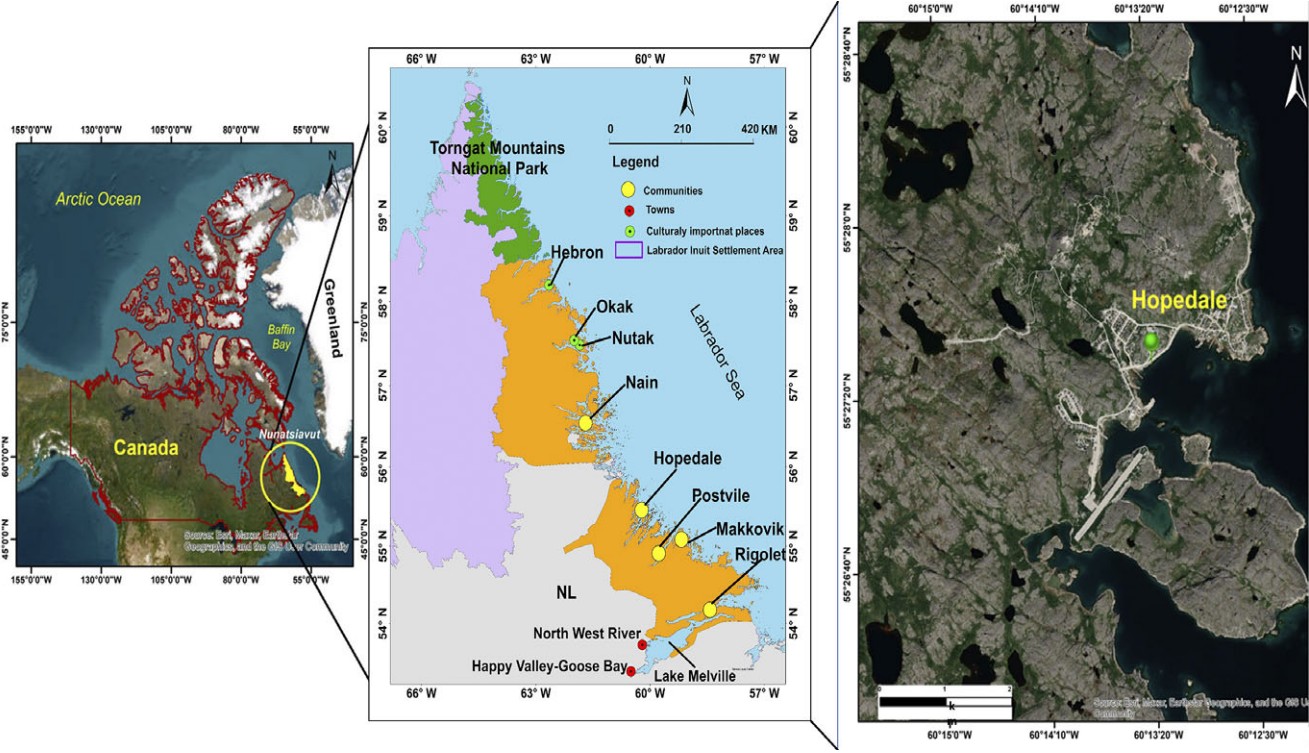

**Figure 1.** Location of Hopedale along with four Inuit communities of Nain, Makkovik, Postville and Rigolet of Nunatsiavut, the Torngat Mountains National Park, Labrador Inuit Settlement Area and culturally keystone places of Nutak, Okak and Hebron in northern Labrador, Canada.

the Nunatsiavut Government. Hopedale holds historical significance due to the forced relocation of many northern Labrador residents from Okak, Nutak and Hebron to Hopedale in 1956 and 1959. Key community infrastructure includes two stores, namely DJ's Convenience Store and Franks General Store, Amos Comenius Memorial School, the Nunatsiavut Government Assembly building, the Hopedale Inuit Community Government office, a hotel named Amaguk Inn (meaning "wolf" in Inuktitut), a community clinic, an airport, the Nanuk Centre for cultural and sports activities, the Inotsiavik Centre for Inuttitut programming – a newly formed non-profit centre aimed to promote cultural well-being, a port and a Moravian church (Figure 2). The Department of Health and Social Development (DHSD) provides essential services and programs related to family services, social development, health and community programs. Hopedale was established in 1782 and was initially named Hoffenthal, meaning "vale of hope" in German, reflecting the first language of many early Moravian missionaries in northern Labrador, while the English version, Hopedale, became more commonly used after 1900 (Nunatsiavut Government, 2019). The missionary station was an important place for Inuit travelling to and from central and southern Labrador for trade (Rollmann, 2013).

## Methods

The research process started with engagement with community members in Hopedale. These discussions focussed on identifying the key themes, issues and community priorities in the context of climate change impacts and adaptation. The key themes identified through this engagement were: how climate change is affecting livelihoods in the community, what aspects are being most affected, what adaptation strategies have been applied by the

community members and what factors are causing constraints in these adaptation strategies. These themes guided the main fieldwork in 2024 in which semi-structured interviews (n = 30), key informant interviews (n = 10) and focus group discussions (n = 5) were conducted with community members. The results were presented in a community workshop at the Nanuk Centre in the summer of 2024 and were also presented to the community members in March 2025 to validate the results and gather feedback from the community. The interviews were transcribed verbatim, followed by thematic analysis. The results were checked, verified, reframed and validated by Inuit community members of Hopedale who are part of this study. The research process, as outlined in Figure 3, encompasses the steps undertaken both prior to and following the execution of the study. These steps were designed to ensure that the research was conducted in a manner that is both respectful and decolonial, thereby addressing and incorporating the concerns and priorities of the community while avoiding knowledge-extractive practices. These decolonial practices help researchers and the communities they study to build two-way relationships, and are particularly important because traditional research methods often reinforce colonial power structures by taking Indigenous knowledge instead of promoting knowledge exchange and community priorities (Omodan, 2025; Joseph et al., 2022). Inuit rely on the land and sea, making culture and daily life highly susceptible to environmental changes. Involving Inuit in research is crucial, as their knowledge and lived experiences can enhance understanding and address environmental change issues in the Arctic more effectively (Furgal et al., 2005).

An active ongoing research collaboration and partnership between the external researchers and the community is in place in which regular visits and contacts are kept with community

**Figure 2.** Hopedale community map outlining settlement types and locations identified as important by community members.

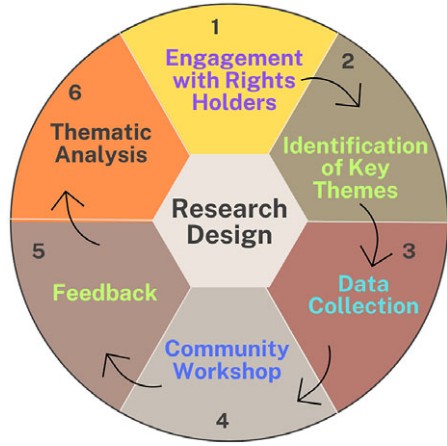

**Figure 3.** Research design describing the different stages of this study.

members to share information and monitor socio-economic and environmental changes. The study was co-designed and conducted in close collaboration with community members, who played a central role in shaping research questions, developing thematic frameworks, contributing to the analysis and co-authoring research outputs.

The data collection process was led collaboratively by Indigenous and non-Indigenous researchers, with significant contributions from community members. Indigenous co-researchers played a central role in shaping the research questions, facilitating interviews and consultations and interpreting findings to ensure that the analysis remained culturally grounded and included community concerns of changing climatic conditions. Thematic analysis was conducted through an iterative process involving both Indigenous and non-Indigenous researchers, ensuring that emergent themes reflected Indigenous knowledge systems.

Semi-structured and key informant interviews were conducted by Indigenous and non-Indigenous researchers, with Indigenous co-researchers leading the engagement process to ensure cultural sensitivity and trust. These interviews were conducted with Indigenous knowledge holders, elders and community leaders identified by Indigenous researchers. Focus groups were facilitated by Inuit researchers to encourage culturally grounded discussions to explore collective community experiences, knowledge-sharing practices and intergenerational perspectives.

The key questions that guided these discussions are

i)   What changes have you seen in weather conditions lately, and how are they affecting the community?
ii)  How do changing sea ice conditions and timing affect harvesting wildlife, fish and firewood?

iii)   Have you experienced any effect on your livelihood due to changing weather conditions?
iv)   Does climate change affect food security?
v)   Are there any traditional foods that you have found particularly difficult to get?
vi)   What adaptation strategies are community members applying to deal with climate change? Are there any factors that affect people's ability to adapt?

These methods were chosen because they allowed for flexibility while ensuring that key themes are explored in depth. The semi-structured format enabled interviewees to share lived experiences without restrictive questioning and allowed flexibility in the conversation. Key informant interviews provided expert perspectives on climate change, decision-making processes and institutional responses. Focus group discussions enabled the documentation of diverse perspectives, shared experiences and adaptation strategies, and the ability to cross-reference and validate them. These are standard methods commonly used in climate change research, allowing flexibility, tailoring research for community priorities, documenting diverse perspectives, changes, experiences, in-depth information and sharing findings and validation of data (Akhter, 2022; Fleming et al., 2022; Belina, 2023; Caggiano and Weber, 2023).

Participants for semi-structured interviews were selected based on their experience with climate change, including hunters, fishers, elders and youth across genders. Key informants were selected based on expertise in climate monitoring, policy or Indigenous governance. Focus group participants included community elders, youth, hunters and women involved in subsistence activities. Five focus groups were selected to capture diverse experiences and intergenerational knowledge from diverse genders and age groups and cross-reference the information shared in semi-structured and key informant interviews. Each group represented a distinct demographic, such as elders with historical environmental knowledge, active hunters and fishers observing real-time ecological changes, women involved in food security and traditional practices, youth experiencing shifts in cultural transmission and community leaders engaged in policy and governance.

Before conducting any semi-structured interview, the information about the study was provided to the participants along with the consent form. Participation in this study was voluntary, and the participants were free to withdraw from the study at any stage. Free, prior and informed consent was obtained before interviews. For key informants, verbal and written consent was obtained, with assurances of confidentiality. All the information collected was anonymised. The interviews were anonymised by using number codes like Participant 1 and Focus Group 1. Culturally appropriate research protocols were followed for focus groups guided by Indigenous researchers, ensuring that culture and Indigenous knowledge were respected.

The research team employed a participatory approach in which Indigenous and non-Indigenous researchers collaboratively coded and categorised data. This was done through the community workshop where key themes were identified and coded. Indigenous co-researchers played a key role in theme development to ensure findings aligned with community perspectives and concerns. The questions were designed to align with Inuit knowledge systems, ensuring that responses were deeply rooted in lived experiences and practical adaptation strategies and constraints.

## Oral history and testimonies

The results are grounded in the oral history and testimonies of community members collected through storytelling sessions from the focus groups to document the community's observations and experiences of adaptations and changing sea ice conditions, seasons, weather patterns and ecological changes impacting livelihoods. Using quotes, we capture these experiences in participants' own words. For example, a community member narrated, "In the last three years, there have been a lot of changes because in July, we usually had floating ice all around us and cold ice weather. We are supposed to be getting more cold winds from the northeast, but it's really hot. Big difference in sea ice from the last few years because I have been ice fishing in our trout ponds all my life for over 30 years. I record ice thickness in my logbook. This spring, the ice was only 1.5 ft thick compared to the 3 ft when I was fishing a few years ago."

We call the approach of using direct quotes in methodology the "Narrative Anchoring Approach." It reflects how direct quotes anchor the methodological framework, capturing the essence of Indigenous knowledge and participatory monitoring as grounded in the personal narratives of participants, voicing community experiences within a methodological structure.

## Indigenous knowledge and participatory monitoring methodological framework

This study is grounded by drawing upon Indigenous knowledge, which is based on accumulated real-time observations of environmental changes. Indigenous knowledge is a cumulative process of long-term, real-time observations, experiences, sharing and intergenerational learning of environmental processes and knowledge (Savo et al., 2016; Reyes-García et al., 2024a). Our approach emphasises that Indigenous observations and experiences provide holistic insights, a multi-causal and culturally grounded understanding of environmental changes and impacts on livelihoods over time (Rapinski et al., 2018; Reyes-García et al., 2024b; Higgins, 2022). By grounding this study in Indigenous knowledge, we ensure that the experiences and observations presented are contextually rich and deeply embedded in local ecological understanding. For example, community members mentioned how the timing of sea ice freezing and melting has changed, affecting traditional hunting routes and livelihoods. As a community member noted, "When I was young, sometimes the harbour would freeze up here in November, but now it is much late, sometimes late December or early January, so we can't travel on it until late." Such testimonies document personal experiences of the impact of climate change grounded in the community's historical and ecological memory.

This study uses a participatory monitoring framework, rooted in Indigenous observational methods, to document and track climate impacts. By engaging community members as active collaborators, this research ensures that monitoring is continuous and responsive to real-time changes. This allows the application of a decolonial lens to challenge the epistemic privilege, violence and authority in Eurocentric or Western knowledge systems (Fanon, 2001). Decolonising research involves fostering what Rauna Kuokkanen, a Sami scholar, calls "multi-epistemic literacy" that promotes learning and dialogue between different epistemic worlds and an ability to read, write, listen, hear and learn, and involves learning as a "participatory reciprocity" (Kuokkanen, 2007; Sundberg, 2014). Regular

discussions and updates with community members were held, allowing for the documentation of changes in real time and fostering shared understanding. This participatory approach promotes knowledge exchange, capacity sharing and co-learning, strengthening diverse epistemic systems.

## Establishing a baseline for future monitoring

This study serves as a baseline from which future changes, impacts and adaptations can be monitored. Documenting past and present environmental conditions and adaptation measures through community monitoring, this study creates a foundation for longitudinal monitoring, enabling future studies to monitor changes in livelihoods, climate risks, socio-ecological changes, coping mechanisms, community resilience and the effectiveness of adaptation measures.

## Positionality statement

This research is a collaborative effort between academic researchers from the University of Leeds (UK), University of Birmingham (UK), Queen's University (Canada) and Inuit community members from Hopedale, Nunatsiavut. The research team consists of 12 authors, including four non-Indigenous researchers affiliated with the University of Leeds (IHM, JDF and DQ) and University of Birmingham (NEB) and eight Inuit community members (IW, BH, NF, KF, MF, DC, CF and RGW) representing diverse genders, backgrounds and expertise.

Indigenous researchers, as community members, played an integral role in shaping the research questions to reflect local concerns, selecting participants and interpreting findings through an Inuit knowledge lens. The external researchers contributed by using established methodologies and analytical frameworks while ensuring that the research remained community-driven. This partnership facilitated a meaningful exchange between Indigenous knowledge systems and academic methodologies, strengthening the study's depth and cultural relevance.

We recognise that power dynamics are inherent in knowledge co-production, particularly in research involving Indigenous communities and academic institutions. This study's research design is participatory and community-led, ensuring that Inuit perspectives are central at every stage. Community members played a central role in determining research priorities, structuring data collection, and approving final interpretations. Co-production of research sought to mitigate the historical imbalances often present in Indigenous research, prioritising local voices over external academic narratives. To ensure that community priorities remained central, multiple measures were taken, such as: (i) community-led governance of research through consultation and decision-making by Indigenous co-researchers, (ii) ethical research practices, including free, prior and informed consent at every stage of data collection and dissemination, (iii) ongoing validation of findings through community feedback sessions, ensuring interpretations accurately reflected lived experiences and (iv) reciprocity and long-term engagement with findings intended to contribute directly to local adaptation efforts and decision-making.

## Ethical considerations

This research was conducted with the consent of community members of Hopedale and adherence to ethical research standards to protect Indigenous knowledge and the rights of participants. All participants were provided with a clear explanation of the study's goals, their rights and the intended use of their data (informed consent). This process followed the principle of free, prior, and informed consent, ensuring that participation was voluntary, with the right to withdraw at any time. Information was conveyed both verbally and in writing to ensure full understanding. Confidentiality protocols were followed, data were anonymised and access was restricted to approved research team members. The study received formal ethical approval from the Nunatsiavut Government Research Advisory Committee (NGRAC-12770416) and the University of Leeds, UK (AREA FREC 2023–0596-660).

## Results

### Climate change impacts in Hopedale

#### Changing Sea ice conditions

Sea ice acts like a highway for transportation and accessibility, facilitating travelling, hunting and fishing and inter- and intracommunity mobility for community members. Over the past decade, communities' observations of sea ice conditions reveal drastic changes in ice dynamics, including variations in freezing and melting processes, the timing of ice formation and changes in its thickness and physical properties. The sea ice is described as becoming unpredictable, thinner, less stable, softer and lasting for a less amount of time over the years.

Community observations reveal that the temporal dynamics of sea ice formation and thawing have undergone significant changes, with ice now forming later in the year and thawing earlier in the spring. This shift has increased the risks associated with travel and resource gathering, consequently affecting the timing of critical activities such as fishing, hunting and wood collection. Typically, sea ice would freeze in November or December or sometimes even in October; however, recent observations indicate that freezing now occurs in December or January. Similarly, the melting period has advanced from June to April or May. A community member narrates, "The sea ice is freezing up later than usual. Normally it would freeze up in early December. Now the first skidoo on ice in 2024 was January 1st, so it's almost a month later." A community member narrated in 2025, "It was mid-January when the sea ice really froze. It froze up in December, but then we had some rain and not much cold weather. That is when we lost most of our snow. So, it stayed like that for almost a month before we got snow again." A key informant mentioned, "In the last 15 years, two specific years stand out: the winter of 2023–2024 and the winter of 2012, when the ocean didn't freeze until mid-January, a departure from normal." Another key informant mentioned, "In 2010, the ocean froze late in mid-January."

Explaining the characteristics of sea ice and its formation, a community member noted, "This winter, pack ice was only maybe between seven to 12 kilometres from town out. Back then, it used to be at least 15 to 30 kilometres out. That is a lot of difference – almost half." A key informant narrated, "When I was a teenager or younger going in the boat with my dad, the ice used to form quicker, which was probably early October or mid-October, and then you would have to put your boat away for the year. Now, take last year, for example, 30 years later, we were in boat until December 29th, and that is a month and a half late."

Community observations have highlighted the early thawing of sea ice, with an elder noting, "Sea ice is thawing out earlier in the springtime than it used to back in the 1980s and 1990s and

definitely in the 1970s." Another respondent reported, "Usually the sea ice would melt in late May and early June, but this year in 2024 it started melting in late April and ended up being thawed out around the 18th of May; the ice was gone in the harbour." A youth narrated their recent experience with sea ice, stating, "Within the last 5 years, dramatic changes have happened in sea ice as it has taken longer to freeze and very much quicker to thaw in the early spring. Usually, even as a kid growing up, I would see the ice safe up until the end of May. This year and last year it was up until the end of April." Another respondent mentioned, "The ice isn't as thick as it usually is, and it's gone a lot sooner than it should be. So, I always go late, like the last weekend in May, to go fishing. You can still get around; there are some parts that were kind of iffy, but now the 24th of May, you can't really go. It's really not safe anymore."

Community members report that the structural integrity of sea ice has diminished as it has become thinner and softer, leading to earlier melting and rendering it increasingly unreliable and unsafe. A community member observed, "At the end of December, we had in between the freezing periods; it caused uncertainty, and some areas that were freezing up were a bit slushy underneath, and the ice wasn't as hard as it was, so it was hard to tell which ice was safe enough to go seal hunting on and which ice was not safe." Another community member reflected on the changes over the past three decades, noting, "In the last 30 years, the ice is becoming thinner and softer, affecting travel. Back then, 30 years ago, the ice could probably get anywhere from 4 to 8 ft thick and run farther out in the ocean. Before, like 30 years ago, the flat sea ice, before you hit the rough ice, used to go out way farther as compared to now." A key informant narrated, "In January 2024, we tested the ice south of here and to our cabin and a couple other places, maybe north. There's no solidity to the ice; it's just soft. The ice is not freezing as hard as it used to."

### Changing weather patterns and seasonal shifts

i) **Temperature pattern:** Recent community climatic observations indicate significant alterations in temperature patterns. Winters and summers are experiencing warmer temperatures, accompanied by increased wind, fog and precipitation. The onset of colder weather is delayed, with early warm spells occurring in spring, and the duration of ice formation is notably shorter. Community members report that historically, winters consistently exhibited freezing conditions from late September or early October, with community observations of temperatures ranging from −10 °C to −40 °C. In contrast, recent winters, while still cold, do not sustain such extreme or prolonged temperatures.

A community member narrated, "We are experiencing more rain in winter, mild temperatures and fog." Another mentioned, "We are used to the cold. And so, for summer now, it is like this is hot for us. This is like hotter than our usual temperatures." A respondent mentioned, "Temperature is affecting the water temperature. This is leading to later freeze-ups and earlier break-ups." A key informant explained, "This year there was less snow, a lot warmer days, more rain in the spring, a longer time for the ice to freeze up, and it was quicker to melt this year; the ice and water opened up earlier this year than all the other years due to hot temperatures." Another respondent narrated, "I find winter is on average warmer than 30 years ago when you would always have minus mid-20s to -30s all the time without windchill. And now we barely hit minus 30 and 40s."

ii) **Snowfall pattern:** Community members suggest snowfall patterns have undergone considerable changes. Previously, substantial snowfall would commence from November or December, facilitating the use of snowmobiles. Now, the quantity of snowfall varies annually and is interpreted as being generally reduced compared to historical levels. Snowfall events are described as less frequent, with diminished snow accumulation and delayed onset. In the past, large snowflakes contributed to rapid snow buildup; however, recent years have seen an increase in wind during snowfall, preventing an even accumulation. Community members report that snow that once persisted until late June, or some patches that remained at high elevations into July, now melts earlier, and there is a noticeable lack of snow cover on ice during winter months. An elder mentioned, "We get less snow than we used to, compared to when I was younger in the 1980s." Another member narrated, "In the last three years, I noticed that the snow is not like it used to be years ago. We used to get a lot of snow when I was a kid; you couldn't go anywhere. Now sometimes we are getting snow in late April and May months, but we normally get that in March. March is what we call a snow month, and you will get bigger snowstorms, but we don't get that anymore."

iii) **Rainfall pattern:** Community observations reveal that rainfall patterns have shifted, with summer and fall experiencing reduced precipitation compared to previous decades. Conversely, winter months are witnessing an increase in rainfall and an increase in fog throughout the year. A community member said, "There's not as much rain as there used to be during the summers." Another member narrated, "We don't get as much rain. Like we don't get hard rain anymore; it might be a shower, but not like it used to be. We have had even drizzle in the winter, which you usually don't get. We have seen rain in winter, which is unusual." A community member noted, "In 2025, I saw drizzle for the first time in March, which is unusual. We got rain thrice this winter, which is abnormal."

iv) **Wind pattern:** Community observations show that wind patterns have intensified, with an increase in overall windiness compared to previous decades. High tides are noted as occurring more frequently, with both higher and lower tides becoming more common. There is a notable shift towards more easterly, southern, southeastern and southwestern winds, which tend to bring warmer weather. Historically, winds were predominantly from the North, Northeast and Northwest; as a community member noted, "North and Northwest winds are not coming so much; now more South and Southeast winds." Another member mentioned, "Easterly wind in April is very rare. But in the last few years, we have seen more and more of it. One time it would be almost unheard of. Now we are experiencing Easterly, Southeast, Southwest and South winds way more now."

v) **Seasonal transition:** Inuit in Nunatsiavut follows six seasons – winter, early spring, spring, summer, early fall and fall (Figure 5). Seasonal transitions are becoming increasingly unpredictable as per community experiences. Early fall and fall are extending in duration, while winter is shortening, and early spring and spring are arriving earlier. Overall, spring and fall are lengthening, whereas winter is becoming shorter, with summers experiencing higher temperatures and slightly longer duration. This shift results in more erratic and less predictable seasonal patterns. A community member noted, "We used to look at the calendar. The first day of spring is March 20th. But in the last few years, we had rain in December, January and February compared to waiting for rain in late March, April or May. So, spring varies. For us, it is when the storm stops, the weather changes, the temperature starts warming up and the snow starts to melt. But in saying that, we have been seeing snow melting in December, January and

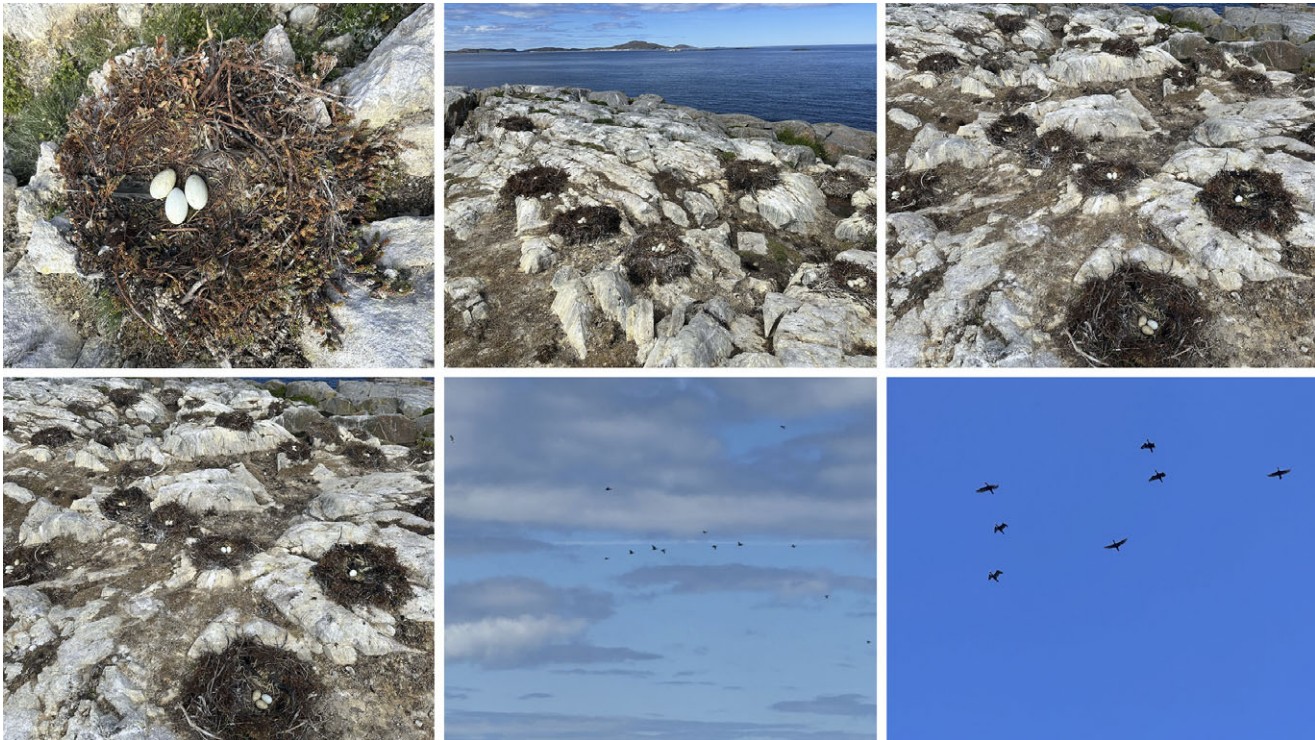

**Figure 4.** Cormorants and their impact on vegetation photographed during fieldwork. A large number of cormorants and their eggs are shown here, and their droppings have affected the vegetation and rocks and turned them white, displacing birds like Eider Ducks and damaging their eggs.

February, which is strange, which are winter months. And we had rain and plus six or plus eight degrees Celsius last year in February."

### Invasive species, new fish and bird species

Recent community observations have documented the appearance of new fish, flies, bugs, bird species and more eagles and songbirds in Hopedale. King salmon have been caught in nets over the past two years, indicating a shift in local aquatic populations. Species such as killer whales and turtles have been sighted, further highlighting changes in marine biodiversity. Conversely, there has been a noticeable decline in the populations of geese and fish. Community observations reveal that geese are migrating south more rapidly, and ice fishing has become less productive. The presence of cormorant birds (Figure 4), which have recently migrated from the south to Hopedale, is particularly concerning. These birds are known to destroy vegetation, turning it white with their droppings, and they disrupt the nests and eggs of eider ducks and other bird species, which are an important food source for the community.

A community member mentioned, "Double-crested Cormorants – we are getting a lot of those now, which we never got before. It is more of an invasive species. It beats off the eggs; it eats a lot of the fish out of the rivers." Another mentioned, "One time, I have seen them in St. Lawrence Seaway; I have never seen them in Labrador. But now they have taken over and are driving out our ducks and other birds that would nest on a horn. They will go on the horn and destroy eggs and everything. So that's one of the birds that I see that we would never see one time." Another member narrated, "And even with the fish, we had sunfish on the north coast last year. Killer whales are moving further north. There are a lot more sightings. One time it would be a rarity." An elder noted, "Plenty of sharks

now. There were always some, but not as plentiful as what there are now, but they disturb the fish that we usually eat. Who's to say in another 20–30 years there will be all kinds of sharks? And they have got to eat something." Another member said, "Summer has been hotter and more flies and bigger ones. Different kinds of animals, insects, flies, bugs and a lot of different birds have been coming up the last couple of years that I have never seen before."

### Community livelihoods under threat

Livelihoods are under threat in Hopedale, particularly for those with fewer resources, due to climate change causing shorter winters and changing ice conditions. These environmental changes significantly hinder wood gathering and hunting activities, which are essential for the community. In particular, the delayed freeze-up restricts access to areas that have traditionally been used for hunting and gathering purposes. For individuals and families who rely heavily on these traditional practices, the difficulties in harvesting and gathering are particularly acute. The increased effort and resources required to obtain firewood and secure food sources place a substantial burden on their economic and social well-being.

The specific impacts of climate change on the livelihoods of Hopedale community members include

### Food (in)security

Climate change has strongly impacted the availability and accessibility of traditional foods as experienced by community members. Warmer temperatures affect the migration patterns and availability of traditional food sources, such as geese and fish, making it increasingly difficult to fish and hunt. Community members have experienced a reduction in the number of animals and the appearance of different species, disrupting ecosystems. This shift not only

affects the availability of these traditional food sources but also challenges the cultural practices associated with hunting and fishing. The altered timing of sea ice freezing and thawing disrupts the timing of activities such as fishing and hunting, which are crucial for the sustenance and cultural practices of the community.

Changes in ice formation and thawing periods significantly impact traditional activities such as food gathering, hunting and travelling. The shorter seasons during which sea ice exists now limit the time available for these essential activities. Climate change is causing animals, fish and birds to migrate farther north, altering the biodiversity and affecting the abundance of species such as partridges and moose. Community members noted that the availability of some berries is adversely affected by the hotter weather. Berries are noted as becoming less plentiful, particularly blackberries, and tend to dry up due to the increased temperatures, further limiting the food resources that the community has traditionally relied upon as food sources and for making jams and cakes. Community members noted that seals are affected by the later formation and earlier melting of sea ice, affecting their number, size and availability.

Climate change has affected traditional food sources like caribou, fish, seals and birds, resulting in a decrease in their number and changing migration patterns. Previously, George River Caribou formed an important part of the Inuit diet, but with their decreasing number, a ban on their hunting since 2013 has meant food has had to be sourced from elsewhere.

### Economic impacts

The economic impacts of climate change are significantly affecting people's livelihoods, resulting in inflation and increasing spending on essential goods and services. One of the most immediate effects is the rising cost of food, which places a substantial burden on the economic well-being of communities. As climate change disrupts traditional food sources, more people are forced to rely on store-bought food, which is becoming increasingly expensive. This shift exacerbates financial strain, particularly for families with limited resources.

Community members report that the cost of building and maintaining infrastructure has risen. While cabins have always been more expensive to construct than tents, increasing material costs and logistical challenges have made them even less affordable. The financial burden of maintaining transportation and communication tools – such as skidoos, All-Terrain Vehicles (ATVs), gas and satellite phones – continues to grow. Although gasoline-powered boats have been widely used in Hopedale (and Nunatsiavut in general) for decades, rising fuel costs and maintenance expenses present ongoing challenges for communities that rely on them for travel and subsistence activities. These technologies, while essential for adapting to changing environmental conditions, require significant investments that many community members struggle to afford. Some community members are unable to afford skidoos, boats and gas necessary for hunting and travelling.

Budget adjustments are also necessary for clothing, as climate change leads to wetter winters and unpredictable weather patterns as experienced by community members. Community members now need to invest more in high-quality waterproof equipment for both spring and fall, as well as better winter clothing to cope with the changing environmental conditions. The increased cost of clothing further strains household budgets, highlighting the broader economic challenges posed by climate change.

### Wood gathering

The challenges of harvesting firewood have intensified due to climate change, necessitating longer travel distances to obtain sufficient supplies. Many community members now resort to purchasing firewood from Goose Bay or local sources. The shorter winters and changing ice conditions complicate wood gathering, disproportionately affecting those with fewer resources. The delayed freeze-up results in wood shortages as access to necessary areas is restricted by water, and transporting wood on boats as compared to skidoos is expensive due to the higher use of gas. The high cost of gas further exacerbates the challenges, prompting more people to rely on wood for heating. Many families are using traditional wood stoves for heating that consume a large amount of wood, creating a high demand for wood collection. However, obtaining firewood without ice is difficult, leading some to source it from coastal boards or ferries for winter storage. The reliance on electricity and furnace heat has increased, despite the high expenses, leaving many with no alternative.

In the past, communities burnt significant amounts of wood, but now the ability to gather wood has diminished. Hopedale, being far from firewood sources, faces additional challenges due to the scarcity of trees in the area. The time once dedicated to hunting is now spent gathering firewood due to changing sea ice conditions, which is a considerable inconvenience. The rising costs of gas and equipment contribute to the financial burden of community members who rely on motorised transport for subsistence activities, including gathering firewood. While technological advancements, such as modern four-stroke snowmobile engines, have significantly improved fuel efficiency – allowing for greater travel range compared to older two-stroke engines – these efficiency gains have been offset by increasing fuel prices. As a result, despite improvements in vehicle performance, the overall cost of maintaining and operating essential equipment continues to place strain on household economies.

### Mental health and well-being

Community experiences showcase that the health impacts of climate change are becoming increasingly evident, manifesting in various physical and mental health issues. Such experiences have also been documented in other communities of Nunatsiavut (Middleton et al., 2020; Sawatzky et al., 2021). Community members have reported an increase in illnesses such as colds, flu and pneumonia, as well as a rise in allergies and asthma among children. Some attribute this trend to changing diets, noting a shift away from traditional wild foods towards store-bought alternatives, which community members believe may impact overall health and resilience.

Hopedale has only one community clinic with limited medical facilities, and doctors typically visit once a month. Weather conditions significantly affect the ability to attend medical appointments outside Hopedale, further impacting overall health. Flight cancellations due to adverse weather conditions leave individuals stranded and unable to get critical medical care, exacerbating health problems.

Mental health is profoundly affected by climate change due to community members' feeling of being landlocked (Table 1). Longer periods of being confined to town due to unsafe ice conditions contribute to enhanced feelings of anxiety and depression. The uncertainty and worry about losing access to traditional foods, witnessing environmental changes and losing the sense of place add to the mental strain. The unpredictability of ice conditions causes significant anxiety and nervousness, especially when family

**Table 1.** Quotes describing climate change impacts on Inuit livelihoods in Hopedale

| |
|---|
| "In the last 20 years in Hopedale, we have experienced warmer and shorter winters, affecting the livelihoods of people because we depend on cold weather to harvest wildlife and firewood. If the ocean isn't freezing, then it becomes harder to obtain wildlife and firewood" – Male elder. |
| "Heat and food are most important. If you can afford to heat your homes and get traditional foods, those are the two most important aspects, but climate change is disrupting this and old ways of life" – Male hunter. |
| "Due to less snow in the last few years, many young ringed seals born in March didn't survive. They lacked the snow shelter (called Aglu) that protects them, leading to high mortality rates" – Female teacher. |
| "A lot of times we only get freight coming because planes can't get here. We are missing lots of days without groceries or freight coming up, causing food shortages" – Female respondent |
| "Livelihoods have been affected due to the shorter period of hunting and gathering. This year, a very few partridges were killed in Hopedale, significantly impacting food supplies. But there is not much available. Because of the melting of sea ice, they are hard to find. We need to make quick hunting trips"– Male hunter. |
| "Going to Torngat Mountains National Park in the north to get Caribou is difficult because it is expensive. There are so many people there, and there is not much space. Caribou has been a part of our diet and culture, but we can't get enough of it" – Youth hunter. |
| "Mental health is affected. We call it the landlock period – a time when the weather conditions are too bad and we get stuck in town for long and can't go having fresh air. Late spring is bad to travel. We are landlocked for a longer time. Usually, we go to cabins because it is nice and quiet, just like the freedom of being away. Last year and this year, we had to suffer and couldn't travel on land" – Key informant. |
| "The changing weather conditions are affecting access to traditional foods, harvesting, seals, and fishing in a big way. Ice fishing in the spring is affected. We can't access places that we would normally go to gather firewood due to late freezing and early melting of sea ice" – Key informant. |
| "Climate change impacts our Canada goose hunt in the spring. We often go hunting for Canada geese, from the end of April and beginning of May up until the end of May. And this year, many hunters, including myself, never had a chance to hunt Canada geese and to go on our traditional hunt for geese in the springtime because the ice was already gone" – Youth hunter. |
| "Before, you would set aside a specific amount because our income is limited up here, right? The cost of living is too high. So, if I put away $50 for clothing, preparing for the weather, but the weather is different now, I have to spend an extra $200, affecting my yearly budget. The clothing I needed before is not adapted to the current weather. It's not in line with the weather anymore, so we need more rain gear now. The rain gear we need is more expensive" – Male respondent. |

**Table 2.** Quotes describing adaptation measures used by Inuit in Hopedale

| |
|---|
| "For the last 32 years, I have been recording weather conditions in my weather journal using my own weather station. For example, I recorded on 11 April 1992: temperature is −10 °C; sky conditions have blizzard; wind is S. N (Strong North); time of day is 7:20 a.m. On 11 April 2024, temperature is −1 °C; sky conditions are sunny; wind is L.S.W (Light South West); time of day is 7:20 a.m." – Male teacher. |
| "We are documenting and testing the ice before travelling on it, focusing on its solidity, thickness, and extent. Despite our lifelong experience with reading ice, snow, and winds, conditions have changed drastically. We are now emphasising the importance of testing ice for safety and educating our community, especially the younger generation, through social media and word of mouth" – Key informant. |
| "I have been growing vegetables for three years now. I grow different kinds of potatoes, radish, peas, onions, lettuce, etc. I also grow them in greenhouses. We have chicken, and we also sell eggs to the local people" – Male teacher. |
| "I had enough potatoes last year; I didn't even have to go to the store. I saved 250 Canadian dollars last fall. What was left over, then I used it for seed this year" – Male vegetable grower. |
| "I use compost to make my own soil and also get soil from land out there. I got seaweed, kelp, and capelin; that is what I use as fertiliser. It is an old-fashioned remedy. It is good for helping the crop grow. I used capelin last year in my tomato pitch, and it had a good return" – Male vegetable grower. |
| "Climate change is changing livelihoods for us, and we have to get seals quicker before they are gone, when the ice melts. Gather wood quicker, get seals quicker. Hunters are adapting to the shorter winter hunting season and try to get as much as possible" – Female respondent. |
| "With the caribou, because we can't hunt them anymore, we are sustaining different animals and moose. More geese, more partridges, more seals, which are sometimes harder to get" – Youth hunter. |
| "Inuit have been historically resilient. It helps us survive. Our traditional knowledge is very important in knowing the land and weather conditions, and food sharing is an important part of our culture that helps us adapt" – Female teacher. |

members venture out on thin ice, highlighting the broader psychological toll of these environmental changes.

### Adaptation

Community members in Hopedale are adapting to the challenges posed by climate change by drawing upon a combination of TEK, social capital, technology, altering wood-gathering activities and diversifying food sources (Table 2). The specific adaptations being practiced by members of the Hopedale community include

### Community-led monitoring and research

Community-based monitoring initiatives are being used to observe and understand changes in environmental conditions, wildlife populations and vegetation and document their effects on local ecosystems and livelihoods. A significant response to changing conditions is the documentation of ice conditions and thickness by community members. Community members record sea ice thickness during the winter and spring months, specifically in April and May. This data collection aims to observe annual variations in sea ice thickness and the amount of snow atop the ice. The process involves drilling holes at various points, from the outer regions to bays and rivers, using devices by some community members such as Conductivity, Temperature and Depth (CTD) Sensors to measure these parameters. The drilling extends to the sea floor to calculate depths and assess water turbidity, with monitoring focusing on understanding freshwater distribution, its impact on river systems and its influence on sea ice formation. Information about sea ice and weather conditions is shared on social media (e.g., Facebook).

### Technology

Community members emphasise the crucial role of technology in adaptation, particularly for navigation and safety, relying on satellite phones (e.g., Garmin satellite phone), GPS (Garmin GPS), satellite messaging devices (ZOLEO and SPOT X), boats and modern snowmobiles, which are designed for durability and extended use in harsh Arctic conditions. Some community members now use iPhones for satellite messaging, which allows for satellite communication in areas without cell signal to send text messages, thereby enhancing connectivity and maintaining communication with the community. Notably, this satellite messaging facility is not available on Android phones. With iPhone

14 models or higher and iOS 18 or higher, community members can send and receive iMessages or SMS messages via satellite when off the grid. Some community members take Wi-Fi devices with them when going to cabins, thereby enhancing connectivity and enabling posting about ice conditions and other updates on social media. These advancements provide reliable transportation over snow and ice, improving mobility and resilience in an increasingly unpredictable environment. Community members are using SmartICE's SIKU maps (https://smartice.org/) for travelling on ice and satellite imagery and weather forecasting for travelling and hunting. These technologies are helping to alter hunting routes to ensure safety and efficiency. Such technologies are used by Inuit in different parts of the Arctic for safe travelling (Bishop et al., 2025).

### Traditional Ecological Knowledge (TEK) and sharing networks

Traditional ecological knowledge (TEK) plays an important role in understanding and adapting to changes, particularly in understanding wildlife migration patterns, assessing the safety of ice and travel routes, determining the timing of seasons, understanding weather conditions and knowledge of traditional foods. This body of knowledge, accumulated and passed down through generations, provides invaluable insights that are critical for survival and sustainability in a changing Arctic environment. Traditional knowledge and practices are also transferred to the younger generation in the school at Hopedale and all age groups at the Inotsiavik Centre for Inuttitut programming (Figure 5). This process involves both classroom instructions and experiential learning, where children are taught various traditional practices and are taken out on the land to gain practical, real-life experience. Inuit resilience, characterised by the historical ability to adapt to harsh and fluctuating conditions, plays a significant role in these adaptation processes. Their resilience is not only a testament to their enduring spirit but

also a critical factor in their capacity to navigate and thrive amidst environmental challenges. Through the integration of traditional knowledge and resilience, Inuit demonstrate a profound ability to adapt, ensuring the continuity of their social fabric, cultural practices and the well-being of the community.

Community members are adapting to earlier thawing of sea ice by harvesting wood earlier in spring and winter and stocking it for use for the rest of the year. In response to the rising costs of gas, a sharing system known as the *"Buddy System"* is being used. This system facilitates the sharing of fuel and machinery expenses among community members for travel and hunting activities. By distributing the financial expenses of gas and the trip, the Buddy System is useful in effectively managing resources and sustaining their communal hunting practices.

There is a noticeable shift towards store-bought foods as difficulties in harvesting traditional foods increase. This shift, while maintaining some food access, does not provide access to preferred cultural foods. Community discussions about climate change have become more frequent, providing comfort and solidarity. Workshops and information sessions are gaining popularity, helping to spread knowledge and foster community resilience, such as the workshop by the Nunatsiavut Government (2024). Community freezers and food sharing among community members are continuing to be important supports for food and cultural beliefs. Food and firewood are shared with community members through community initiatives run by the Nunatsiavut Government and the Hopedale Inuit Community Government.

The observable transition towards store-bought foods, driven in part by the increasing challenges associated with harvesting traditional foods, represents a significant adaptation within food systems. This shift, while facilitating a degree of food security, does not unequivocally translate into a positive outcome. It

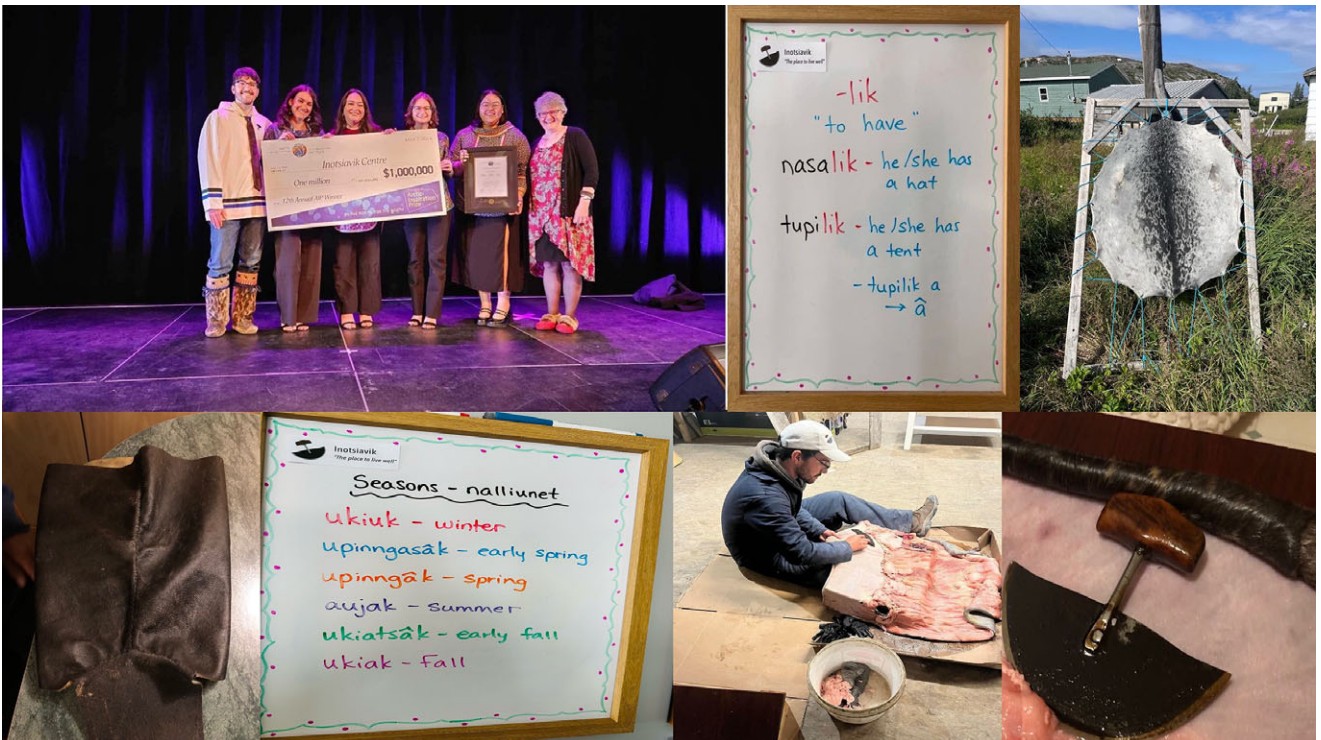

**Figure 5.** Inotsiavik is an Inuit-led initiative dedicated to revitalising Inuttitut language and culture based in Hopedale, Nunatsiavut. As a non-profit organisation, Inotsiavik provides an accessible means to education and programming for Nunatsiavummiut of all ages.

is crucial to acknowledge that such changes, although ensuring the availability of food, often fail to provide access to culturally preferred foods. This phenomenon underscores a complex dynamic where success in maintaining the food supply is achieved at the expense of cultural food practices and preferences. Consequently, the reliance on store-bought foods may lead to the erosion of traditional food knowledge and practices, thereby impacting cultural identity and heritage.

The shift towards store-bought foods, due to increasing difficulties in harvesting traditional foods, represents an adaptation that is therefore not entirely positive. While it helps maintain food availability, it often fails to provide access to culturally preferred foods. This shift can lead to the loss of traditional food practices and cultural identity.

### Agriculture

Agriculture has emerged as an important means of adaptation, and recently, there has been a growing interest in vegetable gardening. Community members are cultivating various crops, including different varieties of potatoes (red, white, purple and Yukon Gold), turnips, peas, tomatoes, radishes, lettuce, romaine lettuce, broccoli, carrots, rhubarb and strawberries (Figure 6). To enhance crop growth, they use natural fertilisers such as seaweed, kelp and capelin. Prior to application, the seaweed is thoroughly washed to remove salt, which could otherwise affect plant growth. Capelin washed on the shores is collected, dried and put in the soil. The vegetables are cultivated on small patches of land. Community members are also constructing boxes and using containers filled with soil to cultivate crops. Besides the naturally available soil, it is also homemade through composting or collected from the surrounding land. Vegetables are also grown in greenhouses. Additionally, some community members are involved in poultry farming, raising chickens and selling eggs at prices lower than those found in stores.

### Challenges to adaptation

Adaptation to climate change in Hopedale is affected by several factors. Some community members struggle to adapt, risking their lives by venturing onto unsafe ice. The high cost of living, machinery and gasoline (for snowmobiles, ATVs and boats), and the limited utility of technologies like snowmobiles due to shorter seasons pose challenges to adaptation. Due to economic constraints, some community members are unable to afford technologies like skidoos or boats to travel and hunt, affecting their livelihoods. Increased dependence on store-bought foods, electricity and furnace heat, which are expensive, further challenges adaptation. Agriculture activities are affected by the limited availability of soil and training and experience in cultivating crops. Changing hunting areas necessitates longer travel, resulting in greater use of gasoline. Travelling to the Torngat Mountains National Park where caribou can be legally harvested is expensive and poses substantial economic burdens on community members. The requirement to cover long distances and dedicate multiple days to these hunting trips leads to increased fuel consumption and accelerated wear and tear on equipment. As a result, this practice is economically untenable for the majority of the community, which also affects the cultural practices and values associated with caribou. Following the ban on caribou hunting, community members turned to moose as an alternative source of meat. However, many members do not find moose as palatable as caribou. Nevertheless, community members report that the moose population has experienced a recent decline, indicating that shifts to new food sources may also have undesirable consequences.

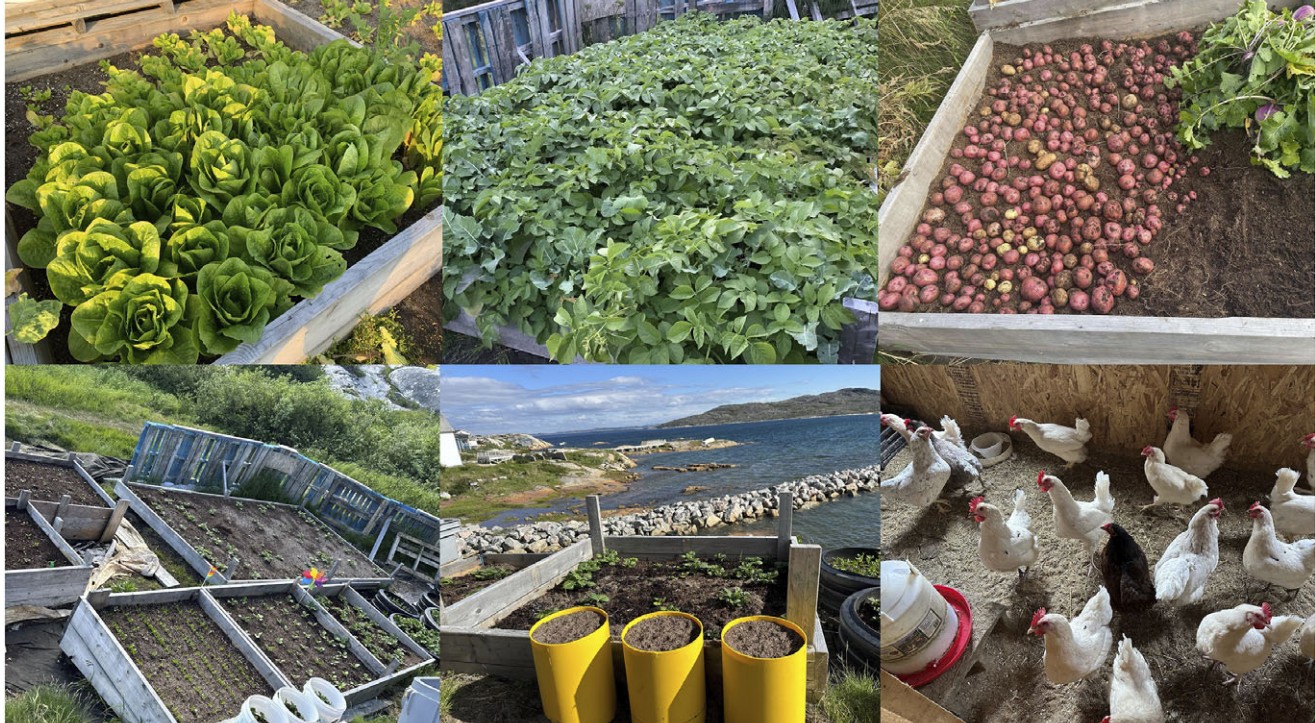

**Figure 6.** Vegetable gardening and chicken farming in Hopedale.

## Discussion

The Arctic is experiencing some of the most rapid climate changes globally, with implications for Indigenous communities, particularly those whose livelihoods rely on sea ice and natural resources (Mardikian and Galani, 2023; Vogel and Bullock, 2021). Community experiences and climate models indicate that Arctic amplification is accelerating warming in the region leading to declining sea ice, increasing air temperatures and shifting precipitation patterns (Taylor et al., 2022; Previdi et al., 2021; Ford et al., 2021). These transformations disrupt Inuit subsistence activities, travel and food security, fundamentally altering traditional ways of life in the Arctic, particularly the Canadian Arctic (Ayeb-Karlsson et al., 2024; van Luijk et al., 2022). In Nunatsiavut, changes in winter trails and travelling on land-based and sea ice trails are affected by changing climatic conditions, creating livelihood changes (Riedlsperger, 2014; Middleton et al., 2020; Wood, 2018). In Hopedale, community observations align with findings in other Arctic communities, where changing sea ice conditions and timing of freezing and thawing affect livelihoods, trail access, transportation and traditional practices that have been documented in Uummannaq, Iqaluit, Nunavut, Nunavik, Foxe Basin, Barrow, Clyde River, Ulukhaktok and Northwest Territories in Canada, Greenland and Alaska (Fleming et al., 2012; Fleming, 2009; Ford et al., 2023; Ayeb-Karlsson et al., 2024; Cooley et al., 2020; Hauser et al., 2021; Hillemann et al., 2023; Baztan et al., 2017; Pearce et al., 2010; Ford et al., 2009). These studies underscore the broader impacts of climate change on Indigenous mobility and safety across the Arctic.

The decline in sea ice extent and stability affects Inuit hunting, fishing and mobility (Ford et al., 2023). In Hopedale, as in other Arctic regions, sea ice has traditionally served as a platform for subsistence hunting, fishing and travel, yet rising temperatures and unpredictable freeze–thaw cycles have shortened the ice season and increased travel risks (Fleming et al., 2012; Fleming, 2009; Chi et al., 2024; Konnov et al., 2022; Wilson et al., 2021). These changes not only reduce hunting success rates but also increase safety concerns for Inuit who rely on ice routes to access remote hunting grounds. Shifts in ice thickness and distribution have affected marine species, with cascading effects on food security and ecosystem balance (Adeniran-Obey and Imoobe, 2024; Pedro et al., 2023).

Climate projections indicate that the Arctic will continue to warm faster than the rest of the world in the 21st century (Lemire-Waite, 2023). The extent of summer sea ice melt will depend on future emission scenarios, with significant implications for ocean heat and freshwater transports into and out of the Arctic (Muilwijk et al., 2024; Wang et al., 2022). These changes could have strong consequences for large-scale oceanic circulation and Indigenous Peoples' livelihoods (Ruiga et al., 2021; Brockington et al., 2023; Maslakov et al., 2022).

Sea ice is of critical importance to Inuit, who have historical and cultural relationships with the marine environment (Aporta et al., 2011). Changing sea ice conditions and weather patterns have profound impacts on Arctic Indigenous traditional hunting and fishing practices, which are crucial for food security and cultural identity (Hossain et al., 2021; Trott and Mulrennan, 2024). The thinning and retreat of sea ice, along with more unpredictable weather, have made travel and hunting more dangerous and less reliable (Raheem et al., 2022). Such patterns are documented in several Indigenous communities in the Arctic by Herman-Mercer et al. (2016), Grigorieva (2024), Kirillina et al. (2023), Charlie et al. (2022), Ksenofontov and Petrov (2024) and Huntington et al. (2022), who found that communities experience increased weather unpredictability and variability, shifting traditional seasonal calendars, changes in precipitation patterns, thinning snow cover, increased temperatures and changing wind directions.

The findings of this study indicate that community livelihoods are under threat in Hopedale, which align with research conducted by Fleming et al., 2012, Fleming, 2009, and in other regions of the Arctic. Indigenous Peoples in the Arctic are experiencing increasing food insecurity, where communities face food shortages, a decline in traditional hunting grounds, subsistence food systems and traditional foods, a decline in wildlife populations and their changing migration patterns, as well as a reduction in berry availability, which is an important part of Inuit diet (Naylor et al., 2021; Huntington et al., 2019; Archer et al., 2017; Konnov et al., 2022; Andronov et al., 2021). Research has shown that climate impacts intersect with multiple socio-economic stresses destabilising Indigenous practices and affecting Indigenous livelihoods, social networks, mental health and overall well-being, hence straining economic conditions (Collings et al., 2016; Fawcett et al., 2018; Lede et al., 2021; MacDonald et al., 2015; Cunsolo Willox et al., 2015). The dependence on expensive store-bought food has increased due to challenges in accessing traditional food sources because of changing ice conditions, making hunting and gathering riskier and less reliable (Wilson et al., 2020; Naylor et al., 2023; Gladun et al., 2021; Fried et al., 2023; Kylli, 2020).

Climate-induced changes to sea ice and weather patterns have exacerbated food security challenges for Inuit communities in the Canadian Arctic (Ayeb-Karlsson et al., 2024). Traditional harvesting of marine mammals, fish and game has become less predictable, requiring hunters to travel farther, adapt to new species availability or shift reliance to expensive, store-bought food (Ross and Mason, 2020; Hillemann et al., 2023; Ford et al., 2021). The increasing reliance on market-based food systems is particularly concerning for Inuit health, given the high costs and lower nutritional value of imported food compared to country foods (Fleming et al., 2012; Fleming, 2009; Little et al., 2021; Malli et al., 2023; Shafiee et al., 2022). Delays in ice freeze-up and earlier melt periods affect the seasonality of hunting, requiring new forms of adaptation that may not always be feasible (Pearce et al., 2021; Dawson et al., 2020).

Climate change affects mental health and well-being in Hopedale. Such experiences have been documented in other communities in Nunatsiavut where the impact on mental health has been linked to the occurrence of ecological grief due to ecological losses caused by environmental change (Cunsolo and Ellis, 2018; Cunsolo Willox et al., 2013; MacDonald et al., 2015; Harper et al., 2015). The intersection of Arctic climate risks with housing shortages, inadequate infrastructure and socio-economic disparities places added pressure on Inuit well-being and self-sufficiency (Rahal, 2024; Alook et al., 2023).

Despite these challenges, Inuit communities in the Canadian Arctic – including Hopedale – have long demonstrated resilience and adaptability in response to environmental changes (Fleming et al., 2012; Fleming, 2009; Vogel and Bullock, 2021; Lede et al., 2021; Ford et al., 2014). This study highlights several community-driven adaptation strategies, including shifts in hunting practices, technological innovations in navigation and the use of Indigenous knowledge to improve climate monitoring. Indigenous Peoples in the Arctic are using GPS technology, satellite imagery and real-time weather data to adapt to changing ice conditions and avoid hazardous routes (Bishop et al., 2025; Tremblay et al., 2018; Naylor et al., 2021). Adaptations being employed in Hopedale are largely consistent with other regions in the Arctic where Indigenous Peoples have traditionally relied on TEK accumulated and

transferred through generations and resilience with the addition of technology and diversifying food resources to cope and adapt to climate change (Nakashima and Krupnik, 2018; Ford et al., 2020; Whyte, 2018; Galappaththi et al., 2019; Mercer et al., 2023). Such adaptations, however necessary, have significant socioeconomic challenges associated with them (Ford et al., 2015; Desjardins et al., 2020; Stepanov et al., 2023).

Inuit adaptation efforts are not without constraints. Limited financial resources, insufficient government support and barriers to knowledge-sharing between generations affect adaptive capacities (Malik and Ford, 2025a). Studies indicate that adaptation planning must go beyond reactive measures and incorporate long-term, culturally grounded strategies that integrate Inuit leadership and sovereignty in decision-making (Hancock et al., 2022; Cadman et al., 2023).

Climate change in Nunatsiavut is altering both the physical environment and the cultural fabric of Inuit communities. Values such as tradition, safety and unity influence how individuals interpret environmental change and determine appropriate adaptation strategies (Wolf et al., 2013). However, conflicting values – such as balancing modern technology with traditional practices – can create barriers to adaptation. Climate variability, including annual and decadal-scale fluctuations, complicates how community members perceive long-term change, making it difficult to separate natural variation from human-induced shifts (Vilá et al., 2022; Finnis et al., 2015). As shrub expansion and wildlife range shifts indicate broader ecological transformations, it becomes clear that adaptation is not solely about responding to climate risks but also about preserving cultural identity and maintaining Indigenous knowledge systems (Whitaker, 2017).

A key constraint to adaptation is the presence of multiple stressors – social, economic and political factors that compound climate risks (Malik and Ford, 2025b; Lede et al., 2021). For example, housing shortages, mental health challenges and systemic marginalisation interact with climate impacts, making it difficult to prioritise adaptation when immediate survival needs are unmet (Bjerregaard et al., 2024; ITK, 2021; Malik et al., 2024). Understanding these cumulative and often compounding pressures is critical for policymakers and researchers aiming to support Inuit-led climate resilience.

## Conclusion

Climate change poses wide-ranging impacts and challenges to Inuit communities, affecting livelihoods and cultural activities. In particular, this study has found that changing sea ice patterns have made hunting, gathering and travelling highly uncertain from year to year and that changing weather has impacted wildlife migration patterns that would normally be a predictable source of food for a large part of the year. The community of Hopedale has successfully implemented several key adaptation strategies, including notably growing subsistence vegetables, establishing sharing networks, using new technologies and monitoring the environment to be able to better understand the timing and rapidity of the changes they experience. Monitoring changes by partnering with Inuit communities can be effective in understanding how climate risks are evolving, how they are experienced and how they shape the lives of community members, as well as the adaptation strategies employed and challenges to them. Inuit in Hopedale exhibit resilience through their experiences and long-held transfer of traditional knowledge and community-sharing practices. Collaborating with

Indigenous Peoples is essential for decolonising research practices and methods and for understanding the intricate relationships between their livelihoods, traditional knowledge and the changing environment. This collaboration fosters mutual respect and integrates Indigenous perspectives, which are crucial for developing sustainable and culturally appropriate solutions to the challenges posed by climate change.

**Open peer review.** To view the open peer review materials for this article, please visit http://doi.org/10.1017/cft.2025.7.

**Data availability statement.** All data is available within the manuscript.

**Acknowledgements.** We are grateful to the community members of Hopedale whose insights and knowledge have been instrumental in shaping this research We thank the Nunatsiavut Government Research Advisory Committee, particularly Michelle Saunders, Carla Pamak, Liz Pijogge and Chaim Andersen for their invaluable guidance and feedback throughout the research process. This study would not have been possible without the significant contributions of our colleagues on the IMAGINE project and without financial support from the Canada-Inuit Nunangat-United Kingdom Arctic Research Programme (CINUK) funded through the United Kingdom Natural Environment Research Council (award NE/X003868/1) and POLAR Canada. We extend our sincere gratitude to Alain Currier for invaluable advice, encouragement and support as we laid the foundation for this research, and whose insights and guidance were instrumental in shaping the direction of this project. We are grateful to Andrew Trant, David Hannah, Luise Hermanutz and Joseph Mallalieu for their insights and guidance. We acknowledge that this research has been conducted on Labrador Inuit Lands and are grateful for this opportunity.

**Author contribution.** Conceptualization: I.H.M., J.D.F.; Data curation: I.H.M.; Formal analysis: I.H.M., J.D.F., N.F., D.Q., N.E.B., R.G.W.; Investigation: I.H.M.; Methodology: I.H.M., J.D.F., I.W., B.H., N.F., K.F., M.F., D.C., C.F., N.E.B., R.G.W.; Original draft: I.H.M., J.D.F.; Review and editing: I.H.M., J.D.F., I.W., B.H., N.F., D.Q., K.F., M.F., D.C., C.F., N.E.B., R.G.W.; Software resources: I.H.M.; Supervision: J.D.F., D.Q., N.E.B., R.G.W.; Validation: I.H.M., J.D.F., I.W., B.H., N.F., D.Q., K.F., M.F., D.C., C.F., N.E.B., R.G.W.

**Financial support.** We acknowledge funding from UK Research and Innovation (Canada-Inuit Nunangat-United Kingdom (CINUK) Arctic Research Programme's IMAGINE Project: NE/X003868/1), UK's Department for Science, Innovation and Technology, Foreign, Commonwealth and Development Office (FCDO) and NERC Arctic Office (ARCWISE Project) and ERC Advanced Grant (via the UKRI Horizon Europe guarantee scheme, EPSRC grant# EP/Z533385/1).

**Competing interests.** The authors declare no competing interests.

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
