## [Reviewer Report]

Thank you for conducting this fascinating work! I thoroughly enjoyed reading your manuscript. However, I have suggestions for improving the manuscript to bring it up to the publishable level.

In lines 112–119, the specific study objectives are unclear. I recommend clearly identifying them. For example: “The three objectives of this study are: 1) xxx, 2) xxx, and 3) xxx.” Providing a concise list will improve the clarity and focus of your study.

In Section 2.2, I would like to see more context about the ongoing research collaborations and partnerships between the researcher (or the team) and the community. Additionally, I have questions such as: Who led the data collection methods? Who was involved in the theme-building and analysis process?

I would like to see more details about the semi-structured interviews, key informant interviews, and focus group discussions. Including following:

• Who uses these methods, and how do researchers apply them?

• What are the guiding questions for each method?

• Why were these methods chosen?

• Who are the participants for each method, and what criteria are used to select them?

• What ethical considerations are associated with each method?

• Why were five focus groups chosen, and in what ways are they different from one another?

• How is thematic analysis conducted, and who is involved in the process?

• How do the listed key guiding questions relate to these data collection methods?

Figure 3 is not cited in the main text, making it unclear how it connects to the narrative described in Section 2.2. Please provide a detailed explanation of its relevance.

Why are oral histories described at the end? Shouldn’t they be included as part of the data collection methods section, alongside interviews and focus groups, at the beginning for better coherence?

Ensure consistency in formatting by using either “quotes” or “quotes” throughout the text.

Please include a concise positionality statement addressing the following: the researchers' roles and relationships with Inuit community of Hopedale, specifying whether they are external collaborators or embedded community members; reflections on how their cultural backgrounds, academic expertise, and institutional affiliations shaped the research design, data collection, and interpretation; acknowledgment of power dynamics in knowledge co-production; and the measures taken to prioritize the community’s voice and ensure its priorities remained central throughout the research process.

Since community members are also part of this study, I would appreciate seeing more detailed information about the ethical considerations for all participants. This could include:

• The process for obtaining informed consent, with specifics on how community members were informed about their rights, the study’s goals, and the intended use of their data.

• Measures taken to ensure confidentiality and data protection, particularly given the sensitive nature of Indigenous knowledge.

• A mention of institutional ethical approval (e.g., from a university ethics board) to demonstrate adherence to established ethical research standards.

Ensure that all figures are cited and thoroughly discussed in the main text.

The current Results and Discussion section covers a range of topics but lacks clear direction. This issue may stem from the absence of clearly defined study objectives at the outset (lines 112–119). By identifying specific objectives, you can better structure your Results section to align with them. Additionally, I am not in favor of combining Results and Discussion into a single section, especially for a case study paper. It would be helpful to understand why you chose this approach.

---

## [Editor Report]

This manuscript does a good job of describing the impacts of changing climatic conditions on the coastal Inuit community of Hopedale. This is clearly a highly relevant manuscript that identifies a range of local adaptation strategies being adopted at the community scale whilst recognising the high value and absolute importance of traditional and place-based knowledges. 

I recommend that the authors carefully consider the suggestions made by the reviewer with regards to a more detailed description of the methodologies employed in data collection and fully explore the contexts in which this research has been undertaken including the suggestion of presenting an authors' positionality statement. This is especially pertinent considering published documents such as the National Inuit Strategy on Research and associated framework for self-determination. A description of the ethical and responsible research considerations that governed this research would be useful especially within the context of power dynamics.

---

## [Editor Report]

The authors must be commended for embracing the reviewer comments so extensively. These efforts have strengthened not only the manuscript but provided a useful mechanism to highlight the deeply-embedded ethical considerations and efforts taken by the author group to ensure cultural sensitivity, trust-building and keeping, and the right to community self-determination throughout the research process. The expanded descriptions surrounding the ethical safeguarding and the methodological decisions made during data collection and analysis are hugely welcome. The inclusion of a positionality statement that clearly recognises existing power dynamics in the co-production of knowledge, and details steps that were taken to address these dynamics, is an important and very worthwhile addition to the manuscript. The diversity of the author group, and inclusion of community members as authors - an action that truly recognises the value of traditional and place-based knowledges - is also to be celebrated.